# Prediction and Verification of Curcumin as a Potential Drug for Inhibition of PDCoV Replication in LLC-PK1 Cells

**DOI:** 10.3390/ijms24065870

**Published:** 2023-03-20

**Authors:** Xuefei Wang, Xue Wang, Jialu Zhang, Qiang Shan, Yaohong Zhu, Chuang Xu, Jiufeng Wang

**Affiliations:** College of Veterinary Medicine, China Agricultural University, No. 2 Yuanmingyuan West Road, Beijing 100193, China

**Keywords:** molecular docking, curcumin, PDCoV, RIG-I

## Abstract

Porcine deltacoronavirus (PDCoV) is an emerging swine enteropathogenic coronavirus (CoV) that causes lethal watery diarrhea in neonatal pigs and poses economic and public health burdens. Currently, there are no effective antiviral agents against PDCoV. Curcumin is the active ingredient extracted from the rhizome of turmeric, which has a potential pharmacological value because it exhibits antiviral properties against several viruses. Here, we described the antiviral effect of curcumin against PDCoV. At first, the potential relationships between the active ingredients and the diarrhea-related targets were predicted through a network pharmacology analysis. Twenty-three nodes and 38 edges were obtained using a PPI analysis of eight compound-targets. The action target genes were closely related to the inflammatory and immune related signaling pathways, such as the TNF signaling pathway, Jak-STAT signaling pathway, and so on. Moreover, IL-6, NR3C2, BCHE and PTGS2 were identified as the most likely targets of curcumin by binding energy and 3D protein-ligand complex analysis. Furthermore, curcumin inhibited PDCoV replication in LLC-PK1 cells at the time of infection in a dose-dependent way. In poly (I:C) pretreated LLC-PK1 cells, PDCoV reduced IFN-β production via the RIG-I pathway to evade the host’s antiviral innate immune response. Meanwhile, curcumin inhibited PDCoV-induced IFN-β secretion by inhibiting the RIG-I pathway and reduced inflammation by inhibiting IRF3 or NF-κB protein expression. Our study provides a potential strategy for the use of curcumin in preventing diarrhea caused by PDCoV in piglets.

## 1. Introduction

Coronaviruses (CoVs) belong to the subfamily coronaviridae and are positive-stranded RNA viruses. From the first coronaviruses, isolated from chickens in 1937 to COVID-19, which is now ravaging the world, the danger of coronaviruses has caused great concern worldwide and poses a serious threat to global public health [1]. The World Health Organization emphasized that early detection was an effective strategy to reduce the spread of CoV virus. Fast and effective testing tools are essential and provide the basis for the later treatment of the disease [2,3]. The increased mutation, recombination rates and high prevalence result in the high genetic variability of CoVs, which further facilitates the spread of CoVs into new hosts [4,5]. The porcine deltacoronavirus (PDCoV) (species name coronavirus HKU15) is a new CoV of uncertain origin isolated from pigs in Hong Kong in 2012 [6]. To date, PDCoV is widely spread in the United States, Canada, China, Korea, Thailand, Vietnam, etc. Neonatal pigs can develop acute watery diarrhea and vomiting due to PDCoV infection of the intestinal epithelium, resulting in dehydration, weight loss and even death, with a mortality rate of 40% [7]. Recent research has found that PDCoV can be transmitted across multiple species using aminopeptidase N (APN) [8,9] and a recent study found the PDCoV strain in plasma samples from three Haitian children with an acute undifferentiated fever in 2021 [10,11]. This finding underlines the potential zoonotic risk of PDCoV as well as it being a human hazard. In light of this, it is necessary to develop therapeutic agents with diverse targets to treat PDCoV infection.

The innate immune response of the host is essential to defend against a viral infection and replication during viral infections. Among the pattern recognition receptors (PRRs), recognition of viral RNA by Toll-like receptors (TLRs) and RIG-I-like receptors (RLRs), inducing the production of type I interferons (IFNs), pro-inflammatory cytokines and activating downstream effectors [12]. The activated RIG interacts with the adapter protein of the mitochondrial antiviral signaling proteins (MAVS), which further recruit the TNF receptor-associated factors (TRAF2/3/5/6). Then, TRAF activates nuclear factor-κB (NF-κB) and interferon regulatory factor (IRF3/7), thus inducing interferon and cytokine production, which can activate downstream antiviral gene transcription [13].

In face of the risks posed by viral infections to humans and animals, the development of novel antiviral compounds encounters several challenges and significant efforts in drug screening and validation. Curcumin (CM) is the main bioactive compound in turmeric; it is one of the most effective nutritional supplements and a Traditional Chinese Medicine (TCM). Turmeric is Generally Recognized As Safe (GRAS) by the US FDA, and curcumin has been granted an acceptable daily intake (ADI) level of three mg/kg-BW by the joint FAO and WHO Expert Committee on Food Additives in 1996 [14]. Curcumin has a potential role in the treatment and prevention of specific diseases, including oxidative and inflammatory conditions, metabolic syndrome, arthritis, anxiety, viral infections and cancer [15]. Previous work has demonstrated that curcumin is an antiviral compound with activity against a variety of viruses, such as HPV, HCV, Zika and chikungunya viruses [16]. In addition, curcumin was found to inhibit SGIV infection, affecting viral particle attachment at different concentrations in vitro and in vivo, as well as modulating cellular immune and inflammatory responses related to the NF-κB signaling pathway [17]. Network pharmacology relies on network theory as well as biology systems to topologically analyze and predict multiple nodes in a targeted system of interconnected molecules. As a result, it exhibits a broader volume of data with excellent confidence [18]. In order to explore the pharmacology of turmeric more extensively and precisely, we predicted the targets of turmeric using network pharmacology and molecular docking. We also evaluated the mechanism of curcumin inhibiting PDCoV replication in vitro. We hope our research may lay a good theoretical foundation for further study on developing new drugs for piglet diarrhea caused by the PDCoV.

## 2. Results

### 2.1. Intersection Target of Curcumaelongae Rhizoma and Diarrhea

A total of 52 effective components of *Curcumaelongae Rhizoma* were initially identified through the databases mentioned in the method. A total of 118 targets were obtained from 52 effective components after high-possibility screening and de-duplication. A total of 607 diarrhea-related targets were acquired by screening the DisGeNET databases. All of the above targets were mapped to the database UniProt and obtained a UniProt ID. Then, their intersection was taken to obtain eight common targets of *Curcumaelongae Rhizoma* and diarrhea, and these sets are the possible targets of curcumin for diarrhea (Figure 1A, Table 1).

### 2.2. PPI Analysis, GeneOntology Function and KEGG Signal Pathway Enrichment Analysis of Common Targets

The PPI network was constructed by importing eight intersecting targets to the STRING database. The PPI network contained 23 nodes and 38 edges (Figure 1B). The average PPI enrichment *p*-value, average local clustering coefficient, and average node degree were 0.00013, 0.589 and 3.3, respectively. To better summarize the specific functions of *Curcumaelongae Rhizoma* in diarrhea, eight common targets were imported into DAVID 6.8 (https://david.ncifcrf.gov/) (accessed on 14 April 2022) for GO function, annotation and KEGG pathway enrichment. The 23 BP, 2 CC and 2 MF terms were found to meet the screening threshold of *p* ≤ 0.05. The results of the GO enrichment analysis indicated that the gene targets are implicated in multiple BPs, such as response to drug, glucocorticoid, negative regulation of cell proliferation, positive regulation of cytosolic calcium ion concentration and so on. The CC was mainly enriched in endoplasmic reticulum lumen and neuron projection. The MF was predominately involved in growth factor activity and cytokine activity (Figure 1C–E). A total of 20 KEGG pathways were found to meet the screening threshold of *p* ≤ 0.1. Genes classified in the pathway analysis were heavily involved in the TNF signaling pathway, Jak-STAT signaling pathway, PI3K-Akt signaling pathway, cancer-related pathways, inflammatory bowel disease (IBD) and so on (Figure 2A, Table 2).

### 2.3. Network Construction of CTPD and Analysis

The KEGG enrichment results showed that the network of CTPD (chemicals-shared target genes-signal pathway-diarrhea) was established and the mechanism of *Curcumaelongae Rhizoma* on diarrhea was systematically explained. There were 70 nodes and 108 edges interacting with other nodes in the network. Then, the Neighborhood Connectivity value of TNP0001 (curcumin) is 5, implying that it is one of the highly relevant active components in turmeric (Figure 2B).

### 2.4. Molecular Docking Analysis

Curcumin might be the key bioactive component for the treatment of diarrhea. Therefore, molecular docking analysis was conducted to calculate the binding energy of curcumin and these targets. The results showed that curcumin can combine to several binding sites of IL-2 (PDBID: 5LQB), IL-6 (PDBID: 4O9H), NR3C2 (PDBID: 2A3I), SLC6A4 (PDBID: 5I6Z), PIK3CG (PDBID: 7MEZ), BCHE (PDBID: 1P0I), PTGS2 (PDBID: 1CUV) and ADRA1A (PDBID: 6K41) proteins, with their docking scores being lower than −3 kcal/mol (Table 3). A lower binding energy reflects a greater binding affinity between the compound and the binding site. Thus, IL-6, NR3C2, BCHE and PTGS2 are the most likely actual targets of curcumin. The docking results were decorated by PyMOL 2.3.0 to show the 3D protein-ligand complex. After the docking process, the best conformation (Figure 3) was selected and the hydrogen bonds were displayed according to the docking results, for example the interaction of IL-2 and curcumin (Figure 3A) showed that the amino acid residues SER60, ASP109 and TRP111 forms hydrogen bond interactions with curcumin.

### 2.5. Curcumin Inhibits PDCoV Replication

First, CCK-8 was used to determine the cytotoxicity of the CM. The relative cell viability was above 100% after treatment with CM at concentrations of 12.5–200 μM for 48 h (Figure 4A), respectively. In order to make the results more intuitive and visual, the results showed that treatment with 200 μM of CM reduced the number of cells stained with crystal violet (Figure 4B). Based on these results, 25, 50 and 100 μM were chosen as nontoxic CM concentrations for the subsequent antiviral assays. Then, LLC-PK1 cells were infected with PDCoV and treated with CM at different concentrations. The Western blotting analysis demonstrated that CM treatment decreased the amount of PDCoV N protein in a dose-dependent manner (Figure 4C). In addition, the inhibitory effects of CM on PDCoV replication were confirmed using virus titration. The viral titers of the infected cells treated with 50 μM or 100 μM CM were decreased to 3 lgTCID50/mL in a dose-dependent manner compared with the infected cells without CM (Figure 4D). Moreover, the decrease in PDCoV titers was also corroborated by virus-specific RNA detection. The cell-associated virus was collected at 24 hpi and viral genomes were quantified using qRT-PCR. A reduction in the PDCoV genomes was observed after treatment with 50 μM and 100 μM of curcumin (Figure 4E). Given that IFN-β is the most important effector molecule that gives immunity after the activation of the innate immune signal, the effects on the IFN-β transcriptional levels were first evaluated. In LLC-PK1 cells, curcumin reduced the expression of IFN-β induced by the PDCoV in a time-dependent manner (Figure 4F). The SI values showed that CM was effective and safe in treating a specific viral infection in vivo (Table 4).

### 2.6. Pretreatment Curcumin Inhibits PDCoV Replication

Time-of-addition experiments on the PDCoV were performed to further understand the mechanism of how curcumin affects viral replication. The LLC-PK1 cells were treated with 50 μM of curcumin at various times before and after infection with the PDCoV. The viral RNA levels were subsequently determined at 24 h after infection with the PDCoV (MOI = 1), after several rounds of viral replication. The PDCoV was sensitive to curcumin; treatment was most effective when performed prior or at the time of infection (Figure 5A). This result suggested that curcumin’s antiviral effect was before or at the time of infection, which meant potentially prior to the onset of viral replication. Therefore, subsequent experiments only explored the effect of 50 μM CM on the PDCoV.

Curcumin affects viral infection at the time of binding/entry to cells because the virus has several rounds of entry, replication and egress within 24 h. To further determine the mechanism of curcumin, the LLC-PK1 cells were infected by the PDCoV at a high MOI of 10 to ensure all cells were infected to prevent multiple rounds of infection. Then, the cells were treated with curcumin before and after infection and were collected at 6 hpi to analyze for viral RNA and viral titers. The PDCoV RNA levels significantly decreased only when cells were treated with curcumin immediately following incubation with the PDCoV (time of addition of 0 h) (Figure 5B). When cells were pretreated with curcumin (time of addition of −4 h), there was no significant decrease in intracellular viral RNA levels. Similarly, the virus titer in the supernatant decreased only at pretreatment (Figure 5C). These results suggested that treatment with curcumin after viral entry into the cells could not inhibit viral replication.

### 2.7. PDCoV Suppressed Poly (I:C) Induced IFN-β Production through RIG-I Pathway

To understand the inhibition of IFN expression by the PDCoV, the molecular mechanism was explored. As shown in Figure 6A, high levels of type I IFN mRNA in the LLC-PK1 cells was induced by poly (I:C) infection. The optimal concentration of poly (I:C) was 35 μg/mL. Generally, the IFN mRNA levels induced by poly (I:C) were affected in virus-infected cells. However, the PDCoV did not reduce the IFN-α level induced by poly (I:C), which suppressed the IFN-β mRNA levels (Figure 6B). Moreover, The Western blot and the qRT-PCR were used to analyze the expression of the many proteins of the RIG-I pathway. The Poly (I:C)-induced cells, incubated with high multiplicity of infection (MOI = 3) of the PDCoV, significantly downregulated the expression of RIG-I and TRAF6 (Figure 6C) and repressed the transcription of RIG-1, IRF7 and IRF3 (Figure 6D–F).

### 2.8. Curcumin Inhibited RIG-I Pathway Induced by PDCoV

To illustrate the role of curcumin, we further analyzed its influence on the expression of RIG-I signal-related key molecules in PDCoV-stimulated. As shown in Figure 7A, the expression of RIG-I signal-related proteins in the LLC-PK1 cells, including RIG-I, TRAF6, NF-κB and IRF3 presented no significant differences after treatment with curcumin at 6 h. Notably, curcumin downregulated the protein expression of RIG-I, TRAF6 and NF-κB in PDCOV infection at 12 h and 24 h. Interestingly, IRF3 did not significantly change following PDCoV infection (Figure 7A). To determine the role of how IRF3 and NF-κB in the RIG-I-pathway mediates IFN-β induction, we examined whether the use of APDC to inhibit NF-κB affects the effect of the PDCoV on IFN-β activation. The results showed that IRF3 significantly decreased after the inhibition of NF-κB, indicated that IRF3 can complement the effect of NF-κB (Figure 7B).

## 3. Discussion

The infection of piglets by the PDCoV can cause vomiting, diarrhea and even death by dehydration, which seriously threats the development of pig farming. In addition, the severe side effects and the rapid viral mutation prevent antiviral drugs from entering the market. Recent studies have indicated that TCM and their extractions, or derivatives, may be promising alternatives for virus treatments. The antiviral effect of established nanomedicines was found to be enhanced by binding specific ligands to the surface of drug-containing nanoparticles to recognize the molecular composition of the target tissues [19]. Perhaps we can use nanotechnology to increase the antiviral capacity of TCM. The primary active component in turmeric is curcumin, and the safety and efficacy of curcumin have been demonstrated. Dietary curcumin supplementation could significantly improve the clinical outcomes, life quality, high-sensitivity C-reactive proteins (hs-CRP) and the erythrocyte-sedimentation rate (ESR) in patients with ulcerative colitis [20]. In this study, a network pharmacology strategy was used to predict the possible pharmacological mechanisms, and in vitro experiments being performed to confirm them. Our study found that turmeric contains 52 multi-target active ingredients, and eight diarrhea-related targets were identified using network pharmacology from them. Through a comprehensive analysis of the active components and targets, curcumin is one of the herbal compounds proven to treat diarrhea. In addition, the result of the in vitro experiments showed that curcumin dose-dependently inhibits PDCoV replication and suppressed the activation of the RIG-I signaling pathway mediated by the PDCoV.

The clinical symptoms of a PDCoV infection in piglets are often complex and acute, making it difficult to find immediate and efficient medications. At present, management of diarrhea in piglets induced by the virus mostly depends on vaccines and anti-viral drugs applied in a clinic are old-fashioned and scarce. Thus, developing novel anti-viral drugs is essential. Therefore, network pharmacology can be used to predict the mechanism of novel drugs quickly and effectively. Our research found that eight common targets were the intersection targets between the potential targets of turmeric and the diarrhea-associated targets, which were identified to build a protein interaction network shown in Figure 1. For example, the predicted and obtained IL-2 and IL-6 have been associated with the pathogenesis of a viral infection. A summary analysis of 182 children hospitalized with COVID-19 had a high proportion of serum interleukin IL-2 and IL-6 [21]. Additionally, there is a strong association between PTGS2 and diarrhea. For instance, the mRNA expression of IL-1β, IL-6 and cyclooxygenase-2 (COX2, PTGS2) was found to be elevated in the ileal mucosa of weaned piglets with diarrhea caused by pathogenic *Escherichia coli* [22]. In rotavirus-infected Caco-2 cells, an increased COX-2 mRNA expression and secreted PGE2 levels were detected [23]. Second, the potential mechanism of turmeric in diarrhea was revealed by a GO and KEGG analysis. An enrichment analysis suggested that the therapeutic effects of turmeric on diarrhea may relate to antiviral, anti-inflammatory, autophagy and apoptosis (Figure 1 and Figure 2). A previous study found that PDCoV infection activates the NF-κB signaling pathway through Toll-like receptors, inducing extreme inflammatory factor expression and leading to body damage [24]. In agreement with this, previous studies have reported that curcumin reduced pro-inflammatory makers (TNF-α, IL-1β, IL-18, IL-2, IL-6, INOS and PTGS2) [25] and curcumin solid dispersions inhibited the activity of BChE by 64% at 100 μM [26]. The Edge Betweenness, Closeness Centrality and Outdegree are the three most direct and important topological parameters in topology [27]. The network of CTPD was constructed to predict the core compound, target gene and mechanism of action of turmeric against diarrhea (Figure 2). In the CTPD network, the curcumin-related data Edge Betweenness of 29 indicates a greater proximity centrality. The Closeness Centrality data and the Outdegree indicates that curcumin is not an isolated node. According to the above three topological data analyses, the oral bioavailability and drug-like properties of the compounds reported in the previous research, considered that curcumin may be the core compound for the treatment of diarrhea. Moreover, the results of the molecular docking also showed that eight target proteins, such as IL-2, IL-6 and PTGS2 could directly bind to curcumin (Figure 3), which suggested that curcumin may exert antiviral effects through these targets. 

To verify the above results, the anti-PDCoV activity of curcumin was assessed, and the result showed that curcumin efficiently inhibited PDCoV replication. Curcumin has been previously published to inhibit viral replication through multiple mechanisms. For instance, curcumin is efficacious in COVID-19 treatment through enhancing the immune response, antioxidants and anti-inflammatories [28,29]. Moreover, Curcumin inhibits HBV replication, which reduces inflammation and fibrosis, limits the progression of hepatitis, reverses hepatic fibrosis and reduces the progression of cirrhosis [30]. The anti-inflammatory properties of curcumin may potentially reduce the severity of HSV-2 infection [31]. Additionally, the viral invasion of host cells occurs with changes in the membrane fluidity, which is necessary for the interaction of viruses with the cells they are going to infect. Curcumin inhibits the classic swine fever virus by modulating FASN, lipid droplets and ATF6 to reduce lipid accumulation [32]. Similar to our results, the antiviral effect of curcumin occurs at the time of viral invasion, probably due to the nature of curcumin as a lipophilic molecule. 

Viruses evade the host’s innate immunity, enhancing virus replication spreading in the host organism rapidly. It has been found that retinoic acid-inducible gene I (RIG-I)-like receptors (RLRs) are able to recognize viral RNA for antiviral action in innate antiviral immunity [33]. However, recent studies have shown that both viral and host-derived RNAs can trigger RLRs activation; therefore, the appropriate activation of RLRs can effectively exert antiviral effects. By contrast, if RIG-I activation is uncontrolled itcan lead to immune dysfunction and the production of a cytokine storm, which can further lead to severe damage to the body [34]. The PDCoV has developed a number of immune-evasion mechanisms to prolong its survival within the host, including evasion of the RIG-I-like receptor recognition, cleavage or degradation of essential innate immune molecules and the blockade of molecular interactions. In our present study, the PDCoV suppressed the expression of the RIG-I signaling pathway proteins induced by poly (I:C). It is one of the mechanisms by which the PDCoV inhibits the IFN-β induction to evade the host’s immune response (Figure 6). Similarly, it has been reported that the PDCoV N protein inhibits the activation of the IFN-β promoter through the RLR signaling pathway [35]. Furthermore, it was found that the PDCoV non-structural protein 5 (Nsp5) antagonizes the type I interferon signaling pathway by catabolizing human NEMO or the cleaving of STAT2 in HEK293T cells [36]. Accumulating evidence suggests that the cytokine storm plays a crucial role in causing fatal pneumonia. The reduction in pro-inflammatory cytokines (TNF-α, IL-6, CCL-2/MCP-1 and CXCL-10/IP-10) contributed to the inhibition of SARS-CoV-2 replication in Vero E6 cells, while abnormal particle morphology of intracellular viral particles was also observed [37]. The PDCoV-induced RIG-I signaling pathway and the NF-κB protein levels were significantly decreased after curcumin treatment, thereby controlling the inflammatory response produced by viral infection. Similar to our results, evidence suggested that curcumin inhibits cell proliferation and causes mitochondria-mediated apoptosis by downregulating the NF-κB signaling pathway and inducing the reactive oxygen species (ROS) [38]. Previous studies have demonstrated that curcumin inhibits the activation of the NF-κB pathway induced by IAV, HBV and human T-cell leukemia virus type I (HTLV-I), etc., [39,40,41].

## 4. Materials and Methods

### 4.1. Identification of Curcumaelongae Rhizoma Compounds

Analysis platforms were used to identify the potential active compounds of *Curcumaelongae Rhizoma*. The target components were predicted using TCMIP (http://www.tcmip.cn/) (accessed on 15 May 2022) and TCMSP (https://old.tcmsp-e.com/) (accessed on 20 May 2022). “Curcumaelongae Rhizoma” was entered as a query to search for the herb name. Target names were obtained from Drugbank (https://go.drugbank.com/) using “Curcumin” as the keyword (accessed on 27 May 2022). The obtained data from the different databases were combined for analysis. Then, all the targets were converted into gene names by the UniProt database (https://www.uniprot.org/) (accessed on 28 May 2022).

### 4.2. Diarrhea-Related Targets Screening

The potential targets of diarrhea were identified by the DisGeNET V7.0 (https://www.disgenet.org/)(accessed on 2 April 2022), which integrates data from expert curated repositories, GWAS catalogues, animal models and the scientific literature. Diarrhea was imported as a keyword and the diarrhea-related disease targets were provided in the database.

### 4.3. Intersecting Target Identification

The intersection of targets between active compounds-targets of and diarrhea-related targets were determined by Venny software (version 2.1; http://bioifogp.cnb.csic.es/tools/venny/index.html) (accessed on 3 April 2022).

### 4.4. Protein-Protein Interaction (PPI) and Enrichment Analysis

The obtained intersection genes were uploaded to STRING11.0 (https://string-db.org/) (accessed on 12 April 2022) to obtain the relationship of PPI, and a threshold interaction score greater than 0.9. Then, the screened intersection genes, and their roles in the signaling pathways were analyzed to explore their functions. A Gene Ontology (GO)-based functional enrichment and annotation tool and the Kyoto Encyclopedia of Genes and Genomes (KEGG) were used in our analysis.

### 4.5. Network Construction

The “compounds-genes-diarrhea-pathway” network was constructed using Cytoscape version 3.9.0. The network nodes with different colors represent four different units. The interconnecting lines illustrate the direct connection between the different units.

### 4.6. Molecular Docking Assessment

Molecular docking was performed using the AutodockTools-1.5.7 to predict the possible docking energies, sites and binding interactions between the protein-ligand, including IL-2, IL-6, NR3C2, SLC6A4, PIK3CG, BCHE, PTGS2 and the ADRA1A molecule. The 2D structures of the CM were obtained from the PubChem database (https://pubchem.ncbi.nlm.nih.gov/) (accessed on 13 April 2022), and all the investigated proteins for docking were gained from the Protein Data Bank database (https://www.rcsb.org/) (accessed on 16 April 2022). The 3D Draw module in the Chem Bio Office software was used to optimize the CM compound structure, and then the format was converted into the MOL2. The protein-ligand structure in the PDBQT file format is necessary for virtual screening. All the bound waters were removed from the protein structure. The torsion bonds of the small molecule ligand CM were selected and defined. Second, a 3D grid box was established using AutoGrid (part of the AutoDock package). The cubic grids encompassed the binding site where the intact ligand was embedded. Finally, AutoDock was used to calculate the binding free energy of the given ligand conformations in the macro-molecular structure. At the end of the docking run, each docking result contained 10 ligand conformations and the best binding site was finally selected.

### 4.7. Cells and Viruses

The LLC-PK1 cells (ATCC CL-101) were cultured in Dulbecco’s Modified Eagle’s Medium Nutrient Mixture F-12 (DMEM-F12) (Gibco, Grand Island, NE, USA) containing 100 U/mL penicillin, 100 mg/mL streptomycin and 10% heat-inactivated fetal bovine serum (FBS) (Gibco, USA). The PDCoV CHN-HN-1601 strain (GenBank accession no: MG832584) used in this study was provided by Professor Hanchun Yang, China Agricultural University.

### 4.8. Cytotoxicity Assay

The cytotoxicity of Curcumin (CM, HPLC ≥ 98% Solarbio, Beijing China) was assessed in vitro using the Cell Counting Kit-8 (CCK-8, DOJINDO, Kumamoto, Japan) according to the manufacturer’s instructions. Briefly, the LLC-PK1 cells were grown to 80–90% confluence in 96-well plates, and then the cells were untreated or treated with different concentrations of the CM (12.5–400 μM) at 37 °C for 48 h. After 48 h treatment, the cells were washed with 0.01 mol/L PBS and incubated with 100 μL DMEM-F12 with 10 μL CCK-8 solution at 37 °C for 2 h. The absorbance was measured with a microplate reader (Model 680 Microplate Reader, BIORAD, Hercules, CA, USA) at 450 nm.

### 4.9. Crystal Violet Staining

The cell proliferation assays were performed using crystal violet staining. Briefly, cells were plated at a density of 1 × 10^5^ cells per well in 6-well plates and either left untreated or treated with different concentrations of CM (12.5–200 μM) in DMEM-F12 at 37 °C for 24 h. After being washed four times with PBS, the cells were then fixed in 4% formaldehyde for 30 min. Then, wells were washed twice with PBS. Additionally, cells were stained with a 0.5% crystal violet solution for 30 min. After washing with PBS, the cells were placed in a dryer for 5 min and photographed.

### 4.10. Antiviral Activity Assay

A CCK-8 assay was performed to evaluate the antiviral activity of CM against PDCoV. The LLC-PK1 cells were inoculated with 100 TCID_50_ PDCoV suspension in DMEM-F12 medium for 2 h at 37 °C. The growth medium containing different concentrations of CM (12.5–400 μM) were then added to the cells. All the cultures were incubated at 37 °C for 72 h. All wells were added to 10 μL of CCK-8 solution and incubated at 37 °C for 2 h. The absorbance was measured at 450 nm. The inhibition rate (%) = [(mean optical density of test − mean optical density of virus controls)/(mean optical density of cell controls − mean optical density of virus controls)] × 100%. The 50% effective concentration (EC_50_) was calculated using a regression analysis, and the selectivity index (SI) was defined as the ratio of the 50% concentration cytotoxicity (CC_50_) to EC_50_. The viral titers were calculated using the Reed–Muench method [42] and expressed as TCID_50_/mL.

### 4.11. RNA Extraction and Reverse Transcriptase Quantitative PCR

The total cellular RNA was extracted using the TRIzol reagent (Invitrogen, CA, USA) according to the manufacturer’s protocol. The cDNA was synthesized by reverse transcription using a miRNA First-Strand cDNA Synthesis SuperMix (TransGen Biotech, Beijing, China) following the manufacturer’s instructions. RT-qPCR was performed using the TB Green™ Premix Ex Taq™ II (Tli RNaseH Plus) (TaKaRa, Dalian China). The expression of cellular glyceraldehyde-3-phosphate dehydrogenase (GAPDH) was quantified as the internal control. The primers used for RT-qPCR are listed in Table 5.

### 4.12. Western Blot

The cells were lysed with radioimmunoprecipitation assay (RIPA) lysis buffer (Solarbio, Beijing China) containing a protease/phosphatase inhibitor cocktail (Cell Signaling Technology, MA, USA). The protein concentration was quantified using a BCA Protein Assay kit (23227, TermoFisher Scientifc, Waltham, MA, USA); the protein was separated using sodium dodecyl sulfate-polyacrylamide gel electrophoresis and then transfered onto polyvinylidene fluoride membranes. After blocking the membranes with 5% skim milk at room temperature for 1.5 h and incubation at 4 °C overnight, different membranes were incubated with the following corresponding primary antibodies: anti-PDCoV N (1:1000, Medgene, South Dakota, USA), anti-RIG-I, anti-IRF3, anti-TRAF6 (1:1000, ProteinTech Group, Chicago, IL, USA), anti-NF-κB (1:1000, Cell Signaling Technology, USA) and anti-β-actin (1:1000, ProteinTech Group, USA). Horseradish peroxidase conjugated to AffiniPure goat anti-mouse IgG (1:5000, ProteinTech Group, USA) or goat anti-rabbit IgG (1:5000, ProteinTech Group, USA) was used as secondary antibodies.

### 4.13. Statistical Analysis

All the experiments were conducted at least three times. The statistical analysis was performed using GraphPad Prism 7 with a one-way ANOVA or a t-test with Bonferroni correction. The Data were presented as mean ± SEM and *p*-value < 0.05 was regarded as statistically significant. 

## 5. Conclusions

In conclusion, we have evidenced the potential mechanism of curcumin for the treatment of PDCoV-induced diarrhea in piglets through Network pharmacology, molecular docking, and in vitro experiments. In addition, the direct effect of curcumin on the RIG-I pathway and the PDCoV were reported for the first time. The antiviral ability and low toxicity of curcumin have promising applications in the treatment of viral piglet diarrhea. Our study lays the theoretical foundation for curcumin to become a new alternative medicine.

## Figures and Tables

**Figure 1 ijms-24-05870-f001:**
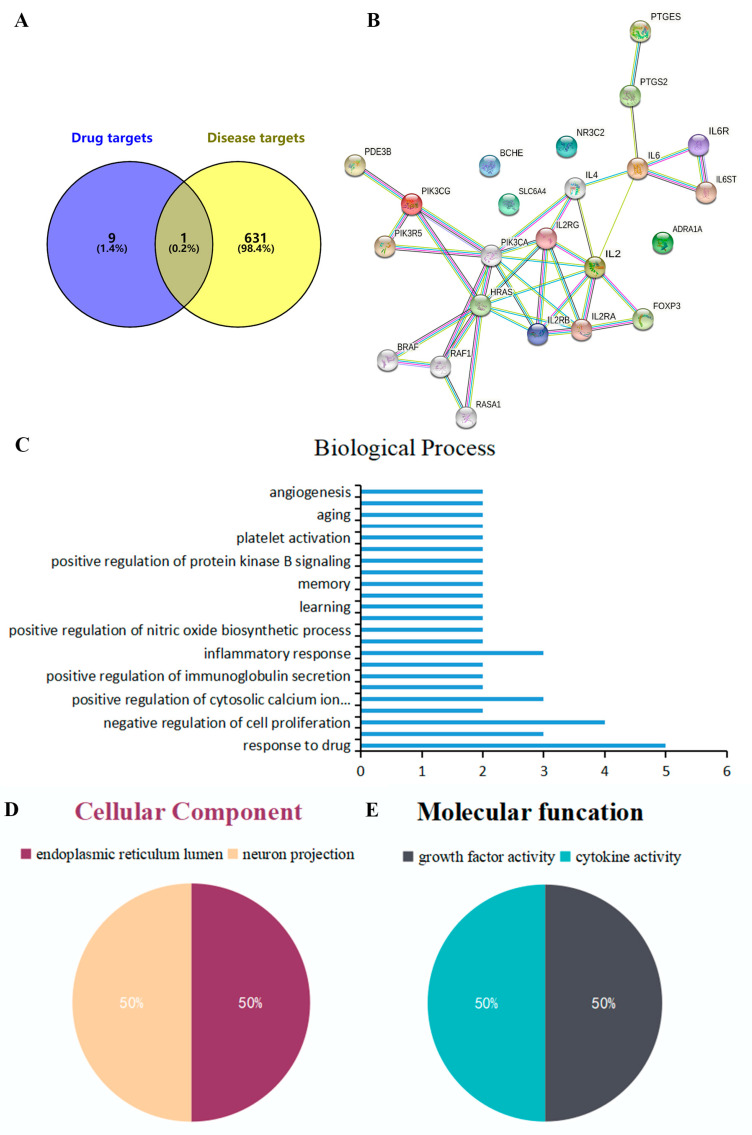
The interaction of shared targets between turmeric active ingredients and diarrhea. (**A**) Venn-diagram of turmeric and diarrhea target genes. (**B**) Protein-protein interaction (PPI) networks. (**C**–**E**) The GO enrichment analysis of the action target genes.

**Figure 2 ijms-24-05870-f002:**
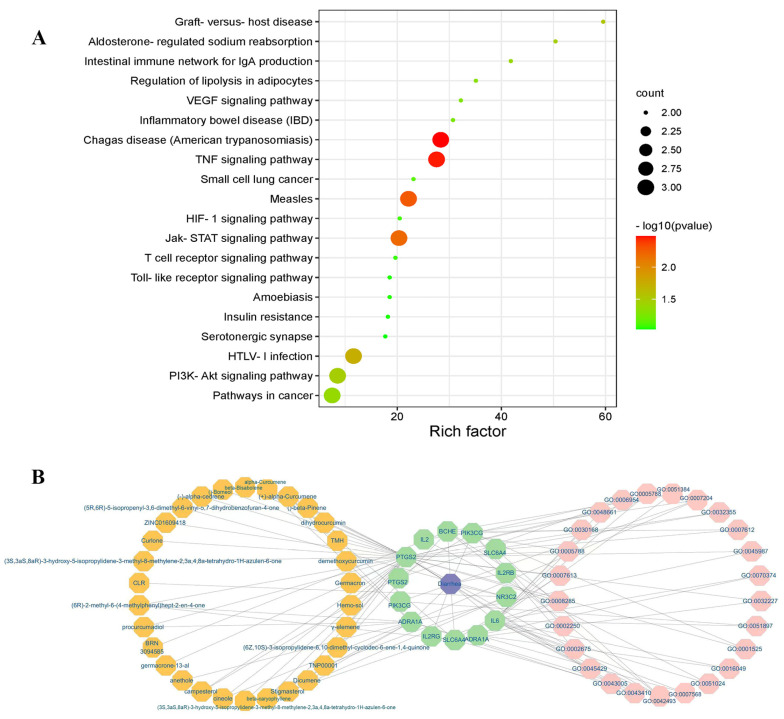
KEGG signal pathway enrichment analysis and the network of CTPD (chemicals-shared target genes-signal pathway-diarrhea). (**A**) KEGG enrichment analysis was performed on intersection targets. (**B**) The pathways are represented in pink font, the bioactive components are represented in yellow font, the common target genes are represented in green font, and the diarrhea are represented in purple font.

**Figure 3 ijms-24-05870-f003:**
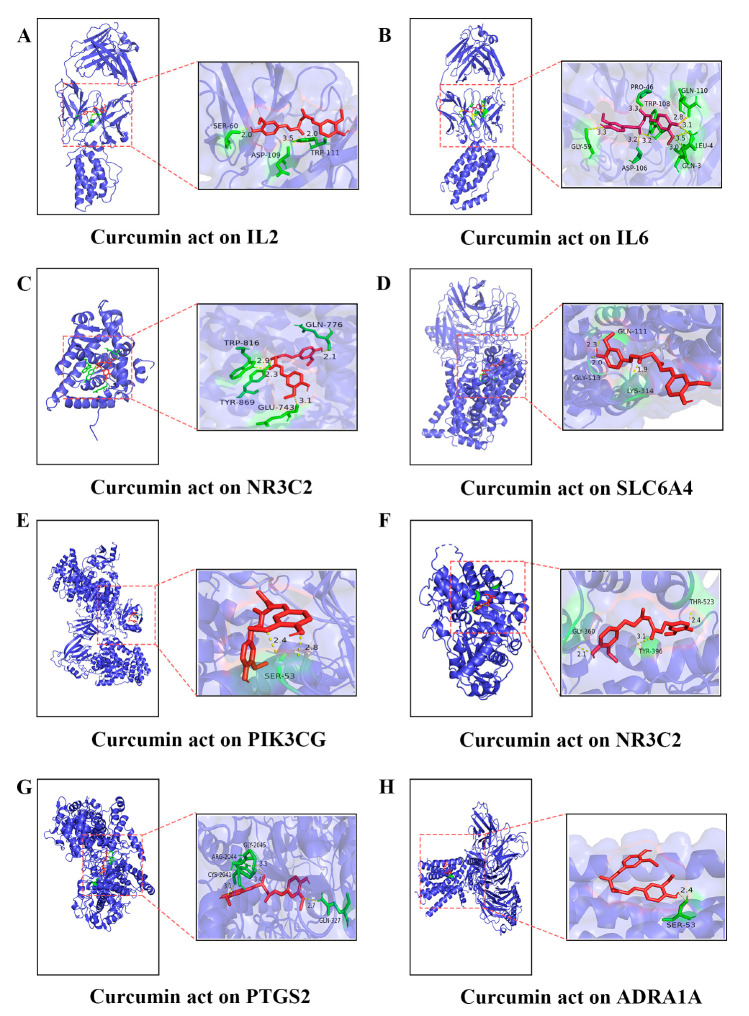
Molecular docking of curcumin with target genes. (**A**–**H**) The molecular docking of curcumin with IL-2, IL-6, NR3C2, SLC6A4, PIK3CG, BCHE, PTGS2 and ADRA1A, respectively.

**Figure 4 ijms-24-05870-f004:**
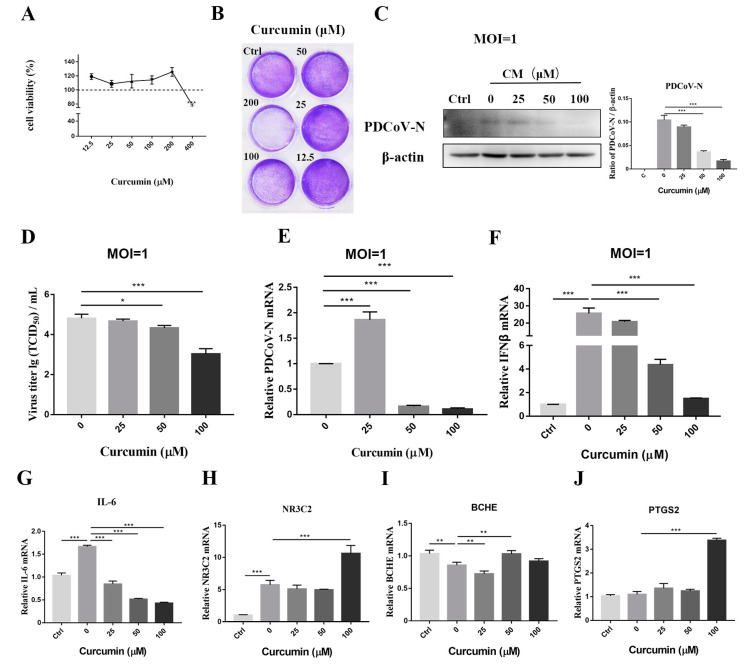
Curcumin inhibits replication of PDCoV. (**A**,**B**) Determination of cytotoxicity of curcumin by CCK8 assay and crystal violet staining. (**C**) The protein levels of PDCoV N in LLC-PK1 cells were determined by Western blotting. Results were presented as the ratio of protein band intensity to the intensity of the β-actin band. (**D**) The viral titer (lgTCID_50_/mL) in LLC-PK1 supernatants was calculated by the method of Reed and Muench. (**E**) The viral RNA copies in LLC-PK1 cells were determined by RT-qPCR with primers targeting the PDCoV N gene. (**F**–**J**) The LLC-PK1 cells were inoculated with PDCoV (MOI = 1) in the presence or absence of curcumin. Total RNA was extracted from cell lysates at 24 hpi. The relative expression of mRNAs was assessed by RT-qPCR. Values represent the mean ± SD for three independent experiments. * *p* < 0.05; ** *p* < 0.01; *** *p* < 0.001.

**Figure 5 ijms-24-05870-f005:**
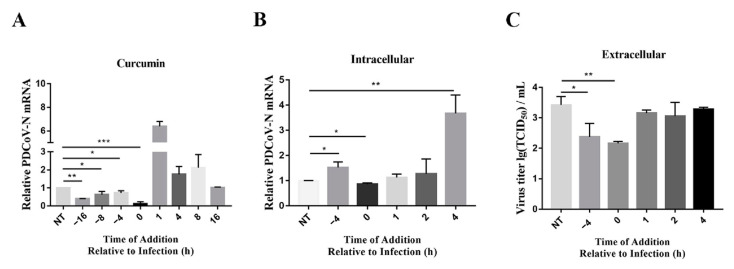
Curcumin inhibits PDCoV with pretreatment. (**A**) Curcumin inhibits PDCoV with pretreatment of cells. LLC-PK1 cells were treated with 50 μM curcumin at the indicated times relative to infection at MOI 1 with PDCoV. Titers were determined at 24 hpi. (**B**,**C**) LLC-PK1 cells were treated with 50 μM curcumin prior to, immediately after, or hours after infection with PDCoV at MOI = 10. At 24 hpi, cells were harvested and intracellular genomes quantified by RT-qPCR or viral titer for PDCoV. * *p* < 0.05; ** *p* < 0.01; *** *p* < 0.001.

**Figure 6 ijms-24-05870-f006:**
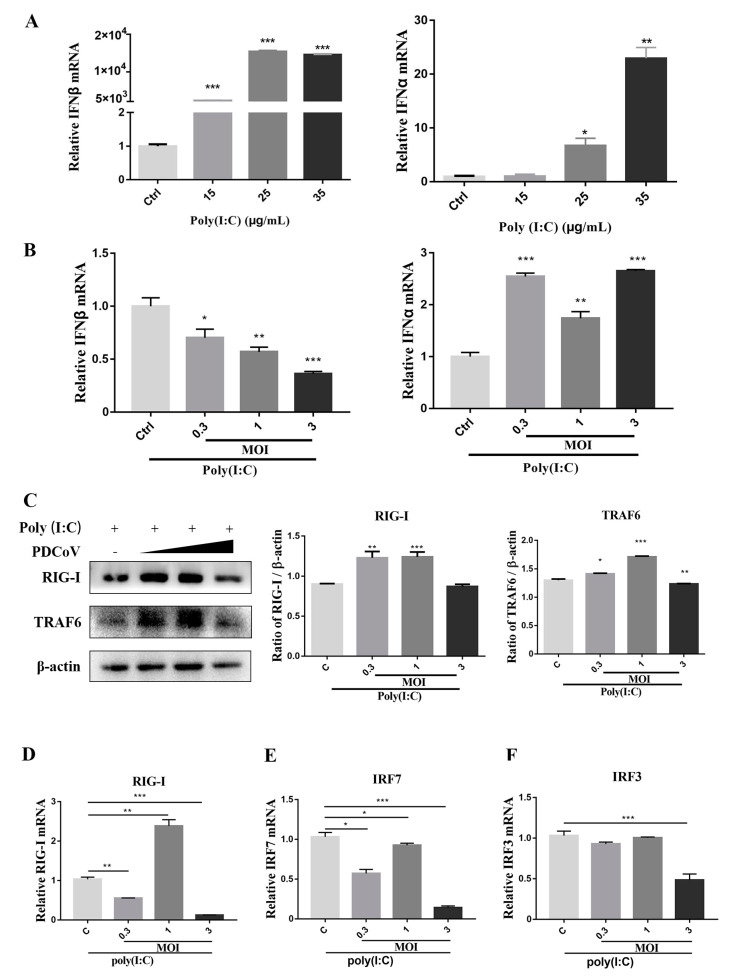
PDCoV suppressed poly (I:C) induced IFN-β production through RIG-I pathway. (**A**) Poly (I:C) induced type I IFN expression. Total RNAs were isolated from LLC-PK1 cells with or without poly (I:C) infection and subjected to RT-qPCR analysis to determine the mRNA levels of IFN-β and IFN-α. (**B**) The effects of PDCoV on IFN mRNA expression. LLC-PK1 cells were infected with PDCoV. At 24 h, the cells were treated with Poly (I:C) for 16 h and then harvested for Western blotting (**B**–**F**) and qRT-PCR analysis. (**C**) The immunoblot signals of RIG-I and TRAF6 proteins were detected. (**D**–**F**) The mRNA levels of RIG-I, IRF3 and IRF7 were quantified following RT-qPCR with specific primers as described in the Section 4. * *p* < 0.05; ** *p* < 0.01; *** *p* < 0.001.

**Figure 7 ijms-24-05870-f007:**
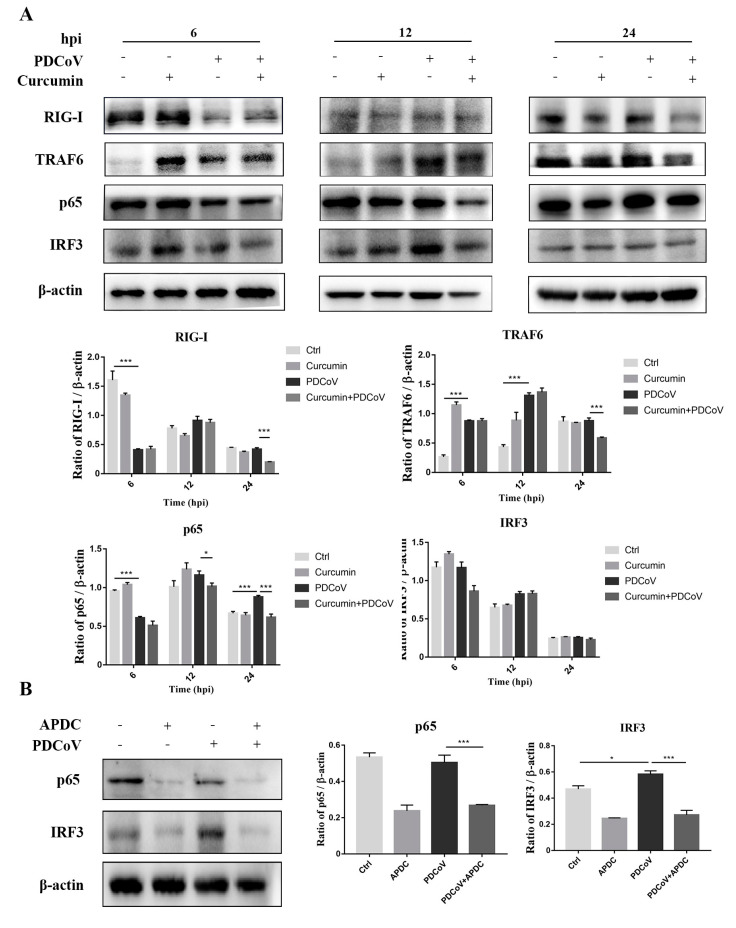
Curcumin inhibited PDCoV-induced IFN-β secretion by attenuating the RIG-I pathway. (**A**) LLC-PK1 cells were inoculated with PDCoV (MOI = 1) in the presence or absence of curcumin. Western blotting analysis of RIG-I, TRAF6, NF-κB and IRF3 was determined at 6 hpi, 12 hpi and 24 hpi. (**B**) The LLC-PK1 cells were pre-treated for 6 h with 150 μM APDC, PDCoV-infected LLC-PK1 cells were examined at 24 hpi. Western blotting analysis of NF-κB and IRF3 was determined at 24 hpi. * *p* < 0.05; *** *p* < 0.001.

**Table 1 ijms-24-05870-t001:** The specific information about the action target genes.

No.	Uniprot ID	Gene Symbol	Gene Name
1	P60568	*IL2*	Interleukin-2
2	P08235	*IL6*	Interleukin-6
3	P35354	*NR3C2*	Mineralocorticoid receptor
4	P35348	*SLC6A4*	Sodium-dependent serotonin transporter
5	P06276	*PIK3CG*	Phosphatidylinositol-4,5-bisphosphate 3-kinase catalytic subunit, gamma isoform
6	P31645	*BCHE*	Cholinesterase
7	P48736	*PTGS2*	Prostaglandin G/H synthase 2
8	P05231	*ADRA1A*	Alpha-1A adrenergic receptor

**Table 2 ijms-24-05870-t002:** The specific information about the Pathways.

No.	Pathway	Count	Percentage %	*p* Value
1	Chagas disease (American trypanosomiasis)	3	37.5	0.003263933
2	TNF signaling pathway	3	37.5	0.00345185
3	Measles	3	37.5	0.005289047
4	Jak-STAT signaling pathway	3	37.5	0.006261011
5	HTLV-I infection	3	37.5	0.018462521
6	Graft-versus-host disease	2	25	0.028450473
7	PI3K-Akt signaling pathway	3	37.5	0.032893784
8	Aldosterone-regulated sodium reabsorption	2	25	0.033550087
9	Intestinal immune network for IgA production	2	25	0.040314858
10	Pathways in cancer	3	37.5	0.041894111
11	Regulation of lipolysis in adipocytes	2	25	0.047878013
12	VEGF signaling pathway	2	25	0.052058258
13	Inflammatory bowel disease (IBD)	2	25	0.054559056
14	Small cell lung cancer	2	25	0.071911173
15	HIF-1 signaling pathway	2	25	0.080893927
16	T cell receptor signaling pathway	2	25	0.084142366
17	Toll-like receptor signaling pathway	2	25	0.088997082
18	Amoebiasis	2	25	0.088997082
19	Insulin resistance	2	25	0.090610547
20	Serotonergic synapse	2	25	0.093026279

**Table 3 ijms-24-05870-t003:** Eight receptor-ligand binding energies.

Receptor	IL2	IL6	NR3C2	SLC6A4	PIK3CG	BCHE	PTGS2	ADRA1A
Binding Energy (KJ/mol)	−19.0	−25.1	−26.4	−15.0	−17.5	−30.1	−27.2	−14.4

**Table 4 ijms-24-05870-t004:** Curcumin SI specific indicators.

Cell Line	Compound	CC_50_	PDCoV	
			EC_50_	SI
LLC-PK1	Curcumin	408 (μM)	5.979 (μM)	68.23

**Table 5 ijms-24-05870-t005:** Primer sequences for rat target genes.

Primers Name	Direction ^a^	Sequence (5′→3′)
PDCoV-N	F	CGCTTAACTCCGCCATCAA
	R	TCTGGTGTAACGCAGCCAGTA
IRF7	F	CTCACCTGCGGTTAACACCT
	R	TTGAAGCCTGGGCCTTCTCC
IRF3	F	GGTGTCTGGCTCAGGAAAGT
	R	AACCGGAAAGAAGCATTGCG
RIG-I	F	GGATGGTAGACAAAGGTGCAGA
	R	GGCTTCAGTGGGCTGTAAGT
IFN-α	F	CCACCTCAGCCAGGACAGAAG
	R	GATGGCATTGCAGCTGAGTAG
IFN-β	F	AGTGCATCCTCCAAATCGCT
	R	GCTCATGGAAAGAGCTGTGGT
GAPDH	F	ACATGGCCTCCAAGGAGTAAGA
	R	GATCGAGTTGGGGCTGTGACT
NR3C2	F	TTCCTCGGCTCGCTTCGC
	R	CCAATGCACGTCACCCAACA
BCHE	F	AGTCCAATTTACAGGCTGGAG
	R	AAGGCTGTTACTGTGCCACC
PTGS2	F	CTGGTGCCTGGTCTGATGAT
	R	TCAATCTGGAAGGCGTCAGG
IL-6	F	GCAGTCACAGAACGAGTGGA
	R	CTCAGGCTGAACTGCAGGAA

^a^ F = forward; R = reverse.

## Data Availability

Not applicable.

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
