# Peer review of "Prediction and Verification of Curcumin as a Potential Drug for Inhibition of PDCoV Replication in LLC-PK1 Cells"

_ijms, 2023, doi:10.3390/ijms24065870_

Round 1

Reviewer 1 Report

The research paper is very interesting and very comprehensive analysis. But, the authors can be enrich the discussion with the potention of virulency of virus if the inflammatory pathway decrase to low. 

Reviewer 2 Report

I would like to recommend this manuscript for publication after minor revision:

1.     The words in Figure 1B can’t be seen clearly. Please enlarge the size.

2.     Figure 2B can’t be seen clearly. Please enlarge the size.

3.     Please carefully check the reference format to make it according to the requirement of MDPI.

4.     There are several references about COV suggested for the Introduction part.

(1) Gang Wang, Le Wang, Zheyi Meng, Xiaolong Su, Chao Jia, Xiaolan Qiao, Shaowu Pan, Yinjun Chen, Yanhua Cheng and Meifang Zhu. Visual Detection of COVID-19 from Materials Aspect. Advanced Fiber Materials, 2022, 4, 1304-1333.

(https://doi.org/10.1007/s42765-022-00179-y)

Reviewer 3 Report

This manuscript deals with " Prediction and Verification of Curcumin as a potential drug for inhibition of PDCoV replication in LLC-PK1 cells ". This article claims that using of  Curcumin could be a suitable for antiviral applications against porcine delta coronavirus . The manuscript is interesting and novel, therefore, I suggest a minor correction and require a detailed clarification. Correction to be addressed by the authors as follows: The abstract is not well organized, where the sentences are incomplete and no continuity is there. It would be feasible, if include the significance of the current study in the abstract. A brief description of how the authors selected information from the literature in the databases, as well as what time period they searched for, is missing. Authors should justify and expand the information on the diseases in which this species is mentioned, highlighting the main contributions. The major text of this manuscript is focused to in vitro  but not in the case of pharmacological activities and clinical applications. Authors should specify the main experimental conditions used on the evidences from the literature. Where they briefly describe the most important data reported in the literature in a homogeneous manner and sequence reinforcing the relevance of curcumin as medicinal alternative. There are various on the application of curcumin based nanoparticle for viral infections, please add new works and disccuss about that. Please discuss about the use of this method for mitochondria targeting in order to best effect of curcumin in cell function. The most significant bioactive metabolites in the effects and in the mechanism of action should be described and noticed more emphatically.  Please add below studies to your manuscript in discussion section:

DOI: 10.1016/j.aquaculture.2022.738870

DOI: 10.2217/nnm-2020-0441

Conclusions should reaffirm the fundamental contribution of this paper.

Reviewer 4 Report

This paper describes the investigators' study of the effects of curcumin in inhibiting the replication of PDCoV. The effort was aimed at investigation of the effect of curcumin on the host reaction to the viral infection and potential interaction of curcumin with the host and much of this has been investigated previously. The investigators seem to be actuing on the assumption that the observed antiviral activity of curcumin is due to virus-host interaction and not a direct effect on the virus. If this is correct then the use of curcumin against PDCoV will be of limited value. The study of curcumin on the host response is interesting but a search for targeting of curcumin against a viral target would be far more useful. The discovery of such a target would likely have greater impact. This study could be published but its impact is questionable in that this is more a study of the host interaction with the agent rather than a study of the interaction of the agent with the pathogen.
